



# Evaluating Three Decades of Precipitation in the Upper Colorado River Basin from a High-Resolution Regional Climate Model

William Rudisill[1], Alejandro Flores[1], and Rosemary Carroll[2]

[1]Boise State University, 1295 W University Dr, Boise, ID 83706
[2]Desert Research Institute, 2215 Raggio Pkwy, Reno, NV 89512

**Correspondence:** williamrudisill@u.boisestate.edu

**Abstract.** Convection permitting Regional Climate Models (RCM) have recently become tractable for applications at multi-decadal timescales. These types of models have tremendous utility for water resource studies, but better characterization of precipitation biases is needed, particularly for water-resource critical mountain regions where precipitation is highly variable in space, observations are sparse, and societal water need is great. This study examines 34 years (1987-2020) of RCM precipitation from the Weather Research and Forecasting model (WRF; V.3.8.1) using Climate Forecast System Reanalysis (CFSR; CFSv2) initial and lateral boundary conditions a 1x1 km innermost grid spacing. The RCM is centered over the Upper Colorado River Basin with a focus on the high-elevation, 750 km$^2$ East River Watershed (ERW) where a variety of high-impact scientific activities are currently ongoing. Precipitation is compared against point observations (NRCS SNOTEL), gridded climate datasets (Newman, Livneh, and PRISM), and Bayesian reconstructions of watershed-mean precipitation conditioned on streamflow and high-resolution snow remote sensing products. We find that the cool-season precipitation percent error between WRF and 23 SNOTEL gauges has a low overall bias ($\hat{x}$=.25%, $s$=13.63%), and that WRF has a higher percent error during the warm season ($\hat{x}$=10.37%, $s$=12.79%)). Warm season bias manifests as a high number of low-precipitation days, though the low resolution or SNOTEL gauges limits some of the conclusions that can be drawn. Regional comparisons between WRF precipitation accumulation and three different gridded datasets show differences on the order of +/- 20%, and particularly at the highest elevations and in keeping with findings from other studies. We find that WRF agrees slightly better with the Bayesian reconstruction of precipitation in the ERW compared to the gridded precipitation datasets, particularly when changing SNOTEL densities are taken into account. The conclusions are that the RCM reasonably captures orographic precipitation in this region, and demonstrates that leveraging additional hydrologic information (streamflow, snow remote sensing data) improves the ability to characterize biases in RCM precipitation fields. Error characteristics reported in this study are essential for leveraging RCM model outputs for studies of past and future climates and water resource applications. The methods developed in this study can be applied to other watersheds and model configurations. Hourly, 1x1 kilometer precipitation and other meteorological outputs from this dataset are publicly available and suitable for a wide variety of applications.





# 1 Introduction

The last decade has demonstrated that non-hydrostatic, convection permitting regional climate models (RCM) are capable tools
for simulating precipitation in complex mountain terrain (Ikeda et al., 2010; Rasmussen et al., 2011; Gutmann et al., 2012; Liu
et al., 2017) and the related task of modeling mountain snow accumulations (Currier et al., 2017; Wrzesien et al., 2019), of
which precipitation in the first-order control. Over 1.6 billion people directly rely on water resources flowing from mountain
regions (Immerzeel et al., 2020), often in the form of seasonal snowpacks. At the same time, mountains are uniquely sensitive
to climate change (Mountain Research Initiative Edw Working Group et al., 2015), and snowpacks are forecast to decline
significantly in the coming decades (Siirila-Woodburn et al., 2021).

However, evaluating biases in precipitation from regional models in a persistent challenge, as gridded, gauge based datasets
that are commonly considered a "gold-standard" can disagree substantially in mountain watersheds because of methodologi-
cal choices alone, and should themselves be considered model products and not treated wholly as observations (Henn et al.,
2018; Lundquist et al., 2019). Gridded gauge based precipitation datasets use various strategies to map sparse gauge obser-
vations across terrain using topographic based regressions (Daly et al., 2008; Thornton et al., 2016). Remote sensing based
precipitation products (such as Ashouri et al. (2015)) are not as suitable for stratiform clouds, nor those composed of ice-phase
hydrometeors, which are both common cases for precipitation in the mountains of the Western United States (Lettenmaier
et al., 2015). Ground based radar systems can measure precipitation rates (Lin and Mitchell, 2005), but radar beam blockage
limits the utility in complex terrain (Maddox et al., 2002). It's fair to say that more work is needed to be done to interrogate
precipitation budgets in mountain watersheds, from RCM or otherwise, as completely adequate observation data are not avail-
able at sufficient scales. Lundquist et al. (2019) articulates this point well and urges the community to consider syntheses of
indirect hydrologic information, such as but not limited to, ecological indicators, soil-moisture, snowpack, and streamflow in
order to better constrain precipitation inputs into mountainous watersheds. The uncertainties in precipitation data (rates, phases,
magnitudes) propagate into studies of hydrologic systems for both water resource applications, snow modeling applications
(Raleigh et al., 2015), and aqueous biogeochemistry (Maina et al., 2020), so improving the quality of precipitation data is of
critical importance in a variety of sectors.

To meet this challenge, this study evaluates 34 years of Weather Research and Forecasting model version 3.8.1 (WRF v3.8.1)
simulated precipitation for a domain encompassing the Upper Colorado River Basin. Special emphasis is placed on the East
River Watershed (ERW), a high elevation $\sim$ 750 km$^2$ watershed in the domain's center where a variety of hydrologic and
atmospheric field campaigns are being conducted. Subsets of the WRF data from this study have been made available publicly
(Rudisill et al., 2022) to provide a baseline of high-resolution climate conditions for scientific activities taking place in the ERW
which include the U.S Department of Energy East River Scientific Focus Area (Hubbard et al., 2018) and SAIL atmospheric
field measurement campaign (Feldman et al., 2021), and the NOAA SPLASH field campaign (https://psl.noaa.gov/splash/).
The Upper Colorado River Basin is currently undergoing a devastating multi-year drought, and this study is timely in so far
that it seeks to provide insight into the quantification of precipitation during the last 30 years across this critical, water-stressed





region (Udall and Overpeck, 2017). The long-term nature of this dataset is also fairly novel, as many studies are conducted for only a handful or years (Ikeda et al., 2010; Rasmussen et al., 2011; Gutmann et al., 2012) or a single decade (Liu et al., 2017).

To begin our model evaluation, we compare our model results against another WRF dataset, namely Liu et al. (2017). This data is publicly available on the NCAR research data archive (Rasmussen and Liu, 2017). This comparison is meant to demon-
strate the fidelity of our model configuration against a well-known and already published data set. We then compare the model against 24 SNOTEL stations (Serreze et al., 1999) in the vicinity of the ERW. Many aspects of precipitation simulation can be important, depending on the question (diurnal cycles, peak intensity, phase, for instance), and not all are considered here, with the focus primarily on seasonal accumulations (Trenberth et al., 2003). We examine the biases in annual, cold-season (October 1 - March 31) and warm-season (April 1 - September 30) precipitation in addition to temporal correlations of accumulation and
daily precipitation rates against SNOTEL data. Lastly, spatial patterns of average precipitation accumulations are compared against three gridded precipitation products, namely the Parameter Regression on Independent Slopes Model (PRISM; (Daly et al., 2008)), Livneh (Livneh et al., 2013), and Newman (Newman et al., 2015) model products. We compare each across the entirety of the WRF inner model grid ($\sim$100,000 km$^2$) and the differences with respect to elevation are considered.

After examining regional-scale precipitation, we focus on evaluating precipitation in the ERW. We examine spatial patterns
of precipitation across the ERW and locations of precipitation enhancement by season. The ERW (Figure 1) is an exemplar of Rocky-Mountain landscapes (Hubbard et al., 2018) and flows from the Elk mountains, approximately in the center of the WRF model domain described in the next section. Elevation ranges between 2500 and 4200 meters above sea level. To better evaluate the differences between WRF and PRISM in the ERW, we compare basin-mean precipitation from each dataset against a Bayesian precipitation methodology. The inference method estimates basin-mean precipitation using a combination of par-
simonious snow/soil water accounting models, precipitation gauge observations, streamflow records, and a limited number of Airborne Snow Observatory snow lidar surveys collected during water-years 2018-2019 (ASO; (Painter et al., 2016)). This work builds upon prior precipitation-from-streamflow work by incorporating lidar-derived snow water equivalent estimates into the precipitation estimating framework, similar to Henn et al. (2016). ASO measures snowpack depths through repeat lidar measurements of snow-covered and snow-free surfaces to produce highly accurate snow depth and snow-water equivalent
maps. The methods developed and reported upon here can be applied to other watersheds and regions. In addition, the precipitation characteristics reported in this study are meant to guide researchers who may use the data for applications and to bring attention to regions where enhanced hydrologic observations prove the most beneficial in the Upper Colorado River Basin.

## 2   Datasets and Methods

### 2.1   WRF Model Domain and Configuration

We use the Weather Research and Forecasting, Version 3.8.1 (WRF; Skamarock et al. (2008); Powers et al. (2017)) model with two nested domains. The inner domain has a 1 km resolution and 50 vertical levels, and the outer domain has a 3km horizontal resolution. The inner grid dimensions is 230 by 236 grid cells, and the outer nest is 349 by 391 grid cells, and the integration timestep is set to 15 seconds. We use CFSR and CFSv2 before/after 2011 (Saha et al., 2010) lateral boundary conditions. CFSR



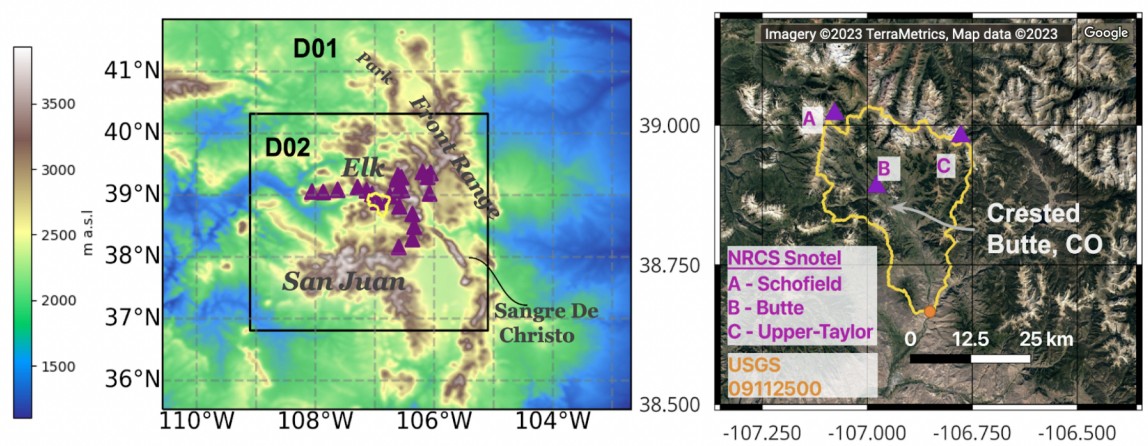

**Figure 1.** WRF Model domains and East River Watershed (ERW) with elevation (left) and satellite imagery (right) displayed. The outer nest (D01) is 3 km dx/dy and the inner domain (D02) is 1 km dx/dy. Purple triangles show the locations of SNOTEL sites examined in this study, and the orange circle shows the USGS Streamgauge at the outlet of the ERW. Major mountain ranges are also labelled.

has a .5 degree horizontal grid resolution. The WRF "namelists" for configuring the model are provided in the supplementary
material. The outermost domain encompasses the majority of Colorado's Rocky mountains, extending East into Kansas and as
far West as the Uintas range. Due to the time and computational constraints, each water-year (October 1 - September 30) is
run independently and preceded by a two week spinup period. Consequently, multi-year soil-moisture/atmosphere interactions
might not be well represented, as the soil moisture fields (and other land surface states) are initialized at the beginning of each
water year with the coarse CFSR soil moisture field. In this way, multiple water years can be run concurrently. The horizontal
grid resolution of both domains is less than the 4km typically considered necessary to permit convection (Weisman et al., 1997)
so convective parameterizations are turned off. Additional WRF parameters are listed in Table 1.

In general, two predominant synoptic regimes control water-inputs to the ERW, namely winter baroclinic waves (frontal systems) and summer-time convective precipitation events that can sometimes be associated with the North American monsoon.
Upper level winds and moisture are predominantly from the West during the winter. The Colorado front range is also affected
by upslope storms typified by Northerly and Easterly winds (Rasmussen et al., 1995). Streamflow hydrographs in the ERW are
typified by a large single peaks during the spring/early summer, occasional summer spikes depending on monsoonal precipitation and antecedent snow conditions, and gradually decays to baseflows in the late summer/fall (Carroll et al., 2020). The
Colorado river system is undergoing a multi-decadal drought primarily driven by increased temperatures (Udall and Overpeck,
2017).





| Grid Options | Outer/Inner Nest | |
| --- | --- | --- |
| WRF Version | 3.8.1 | |
| Vertical Levels | 50, 50 | |
| W-E Dimension | 340, 349 | |
| N-S Dimension | 290, 328 | |
| DX | 3 km,1 km | |
| DY | 3 km,1 km | |
| Timestep | 15 s | |
| Model Physics | Option | |
| Convection Parameterization | None | |
| LBCs | (CFSR;CFSv2) | Saha et al. (2010) |
| Microphysics | Thompson | Thompson et al. (2008) |
| LSM | Noah MP | Niu et al. (2011) |
| Surface Layer | Monin-Obukhov (Option 2) | Monin and Obukhov (1954) |
| PBL | Mellor-Yamada-Janjic (Eta/NMM) PBL | Janić (2001) |
| LW Radiation | Community Atmosphere Model | Neale et al. (2010) |
| SW Radiation | Community Atmosphere Model | Neale et al. (2010) |

**Table 1.** Weather Research and Forecasting v3.8.1 (WRF) parameters used in this study.

## 2.2 Comparison Precipitation Datasets

We compare WRF precipitation accumulations against NRCS SNOTEL precipitation observations. Analysis is conducted by water-year (WY), or October 1 - September 30. SNOTEL stations are designed to provide cost-effective climate information for water-resource important regions throughout the Western US and have been used extensively in the study of hydrology and climate (Serreze et al., 1999). SNOTEL stations use both a snow-pillow to measure snow accumulation mass and a separate, collocated precipitation gauge to measure precipitation liquid equivalent. Ultimately twenty-three SNOTEL sites are compared, ranging between 2400-3350 meters above sea level (purple triangles in Figure 1). The CFSR reanalyses used to force WRF do not assimilate SNOTEL precipitation data (Saha et al., 2010), so precipitation recorded at the SNOTEL station is a completely independent check of WRF precipitation at that grid cell. It is worth noting that there is a fundamental and unavoidable scale mismatch in making such a comparison, given that WRF grid cells are 1 km, and SNOTEL gauge orifices are less than a meter in diameter. Nevertheless this comparison is the best available for quantifying model precipitation performance and has been used as a benchmark in many other studies.

We also compare WRF precipitation fields against the Parameter Regression on Independent Slopes model (PRISM; (Daly et al., 2008)), Livneh (Livneh et al., 2013), and Newman (Newman et al., 2015) geostatistical products, respectively. There are a number of differences between each model product, and elucidating the precise nature of the differences is beyond the





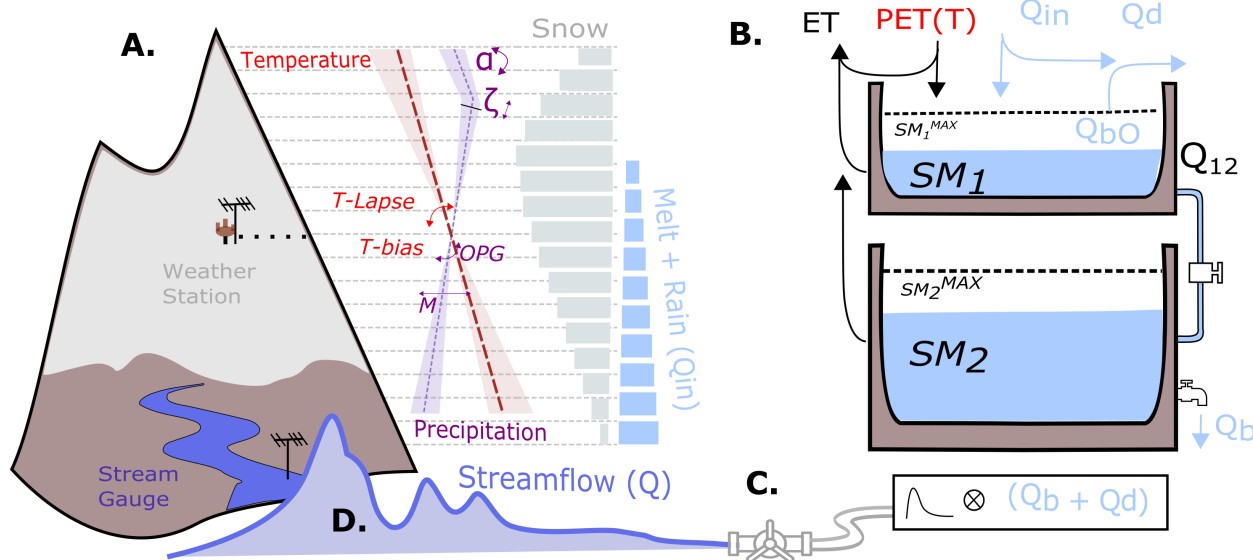

**Figure 2.** Conceptual diagram illustrating the model used for the Bayesian inference approach. (A) Precipitation/temperature increase/decrease with elevation and snow accumulates/melts based on temperature in each layer. Rain and Snowmelt ($Q_{in}$) and Potential Evapotranspiration (PET) calculated from temperature are fed into the hydrologic bucket model (B) with soil moisture in the top and bottom bucket respectively ($SM_1$, $SM_2$) as state variables. Baseflow ($Q_b$) and overland flow ($Q_d$) are convolved with a routing kernel to produce streamflow (D) that can be compared to observations. Bayesian methods estimate precipitation distributing parameters

scope of this article but a brief description is warranted. One key difference is that PRISM and Newman use data from NRCS
SNOTEL networks, whereas Livneh uses observations from the NWS COOP stations that have at least 20 years of data.
Livneh precipitation accumulations are scaled such that the monthly means match mean PRISM climatology from 1961-1990.
All three products use the PRISM terrain-precipitation relationships to distribute precipitation across terrain. PRISM uses a
mapping methodology that regresses precipitation for each individual grid cell based on nearby station observations and terrain
orientation with respect to climatic variables. Livneh and Newman are available until 2012 and 2013 respectively, whereas
PRISM is available for the entire study period. It is also important and necessary to note the differences in grid resolution
among the products. PRISM uses a 4x4 km grid, the Livneh dataset is 1/16° (∼ 6 km), and the Newman dataset is the coarsest
at 1/8° (∼ 12 km). To compare the products at the regional scale, we use a bi-linear interpolation as implemented in the xESMF
(https://xesmf.readthedocs.io/en/latest/) python package. WRF and each respective product is regridded to a common regular
lat/lon grid matching the coarser product's resolution with extents matching the WRF domain. A conservative regridding option
was also tested, but we found that the results were very similar and had little impact on the overall interpretation.





## 2.3 Bayesian Reconstructions of ERW Mean Precipitation From Streamflow

Estimating precipitation from streamflow observations, or "doing hydrology backwards" methods, have been employed in a
number of studies (Kirchner, 2009; Pan and Wood, 2013) including those in snow-dominated alpine watersheds (Le Moine
et al., 2015; Valery et al., 2009) and glaciated watersheds (Immerzeel et al., 2012) using glacier mass-balance as opposed
to streamflow. Henn et al. (2015) used a Bayesian inference method to evaluate gauge-based precipitation products, with
further applications in Henn et al. (2016), Henn et al. (2018), and Hughes et al. (2020), the latter of which used such a
methodology to evaluate atmospheric model performance for watersheds in the Sierra Nevada. The approach adopted in this
study is intended to follow Henn et al. (2015) as closely as possible. In essence, the method combines a temperature index
snow accumulation/ablation model run in elevation bands, a soil-water accounting and streamflow routing "bucket" model, and
precipitation and temperature equations that distribute weather station observations upwards and downwards across elevation
bands. Precipitation tends to increase with elevation, and temperature tends to decrease. A Bayesian inverse method finds the
most likely ranges of parameters, including parameters in the precipitation/temperature distributing functions, that produces
streamflow which best matches observations. The precipitation in each elevation layer (at height z) is given by the following
equation:

$$P_{(z)} = 10^m * [(P_s + P_{bias}) * (1 + OPG * dz)] \tag{1}$$

Where $P_s$ is the daily observed precipitation at a SNOTEL location (at height $z_0$), $P_{bias}$ is a precipitation gauge undercatch
factor, $OPG$ is an orographic precipitation enhancement factor, and $m$ is a multiplicative error term, and $dz = z_{eff} - z_0$
where $z_0$ is the station elevation. Observations show that snow-water equivalent often decreases at high elevations in mountain
watersheds (Kirchner et al. (2014)) after increasing fairly linearly. This pattern is also seen in the ERW (see results section)
and may be related to enhanced sublimation near ridges or a decrease in precipitation. Regardless of the cause, we introduce a
new but simple model that can account for this using phenomena an "effective" layer elevation ($z_{eff}$) which is proposed as:

$$z_{eff} = \begin{cases} z - \alpha(z - \zeta) & \text{if } z > \zeta \\ z & \text{otherwise} \end{cases} \tag{2}$$

Where $z$ is the height of the elevation layer. In this way, precipitation begins to decrease after a certain elevation $\zeta$ with the
slope defined by $\alpha$. This relationship is graphically illustrated in Figure 2. Other non-linear forms of the precipitation/elevation
equation are frequently used (Liston and Elder, 2006), but we chose this novel, linear form for simplicity and because it is
found to well-match ASO data. We use the SNOW-17 snow model (Anderson, 1973) run in discrete elevation layers to simulate
the accumulation/melting of snow. In order to provide estimates that are as independent as possible of WRF, we use NRCS
SNOTEL data from the Butte station located in the ERW to force the model. Periods of missing or poor quality temperature
data (a small percentage) are corrected using adjusted data from the Schofield station or interpolated between neighboring
values. To infer precipitation, a three-part inference process is applied. In the first step, SNOW-17 parameters are calibrated





for the Butte SNOTEL (380) site, including a precipitation undercatch term using an error minimization algorithm. Next, the $OPG$, $\zeta$, and $\alpha$, and temperature lapse rate parameters are fitted to the mean Airborne Snow Observatory SWE for water years 2018 and 2019. ASO produces three-meter scale estimates of snow-water equivalent by taking repeated lidar observations of

snow surfaces and modeling snow density using energy balance modeling. The spatial accuracy of ASO snow depths are on the order of centimeters (Deems et al., 2013). While ASO data presents only a single snapshot in time, the spatial resolution and accuracy are very high and thus contain unparalleled information about the spatial distribution of orographic snowfall (Vögeli et al., 2016), acknowledging that other factors (avalanches, wind redistribution, sublimation) also control the snowpack depth prior to melt. The target ASO dates used are taken at our near peak snow accumulation (early spring), so we expect that prior

melt out is fairly minimal. Since we are using data averaged in elevation bands, wind redistribution effects are also likely small.

| Fluxes | Description | Functional Form |
|--------|-------------|-----------------|
| $Q_{12}$ | Percolation | $ku*(SM_1/SM_1^{max})^c$ |
| $Q_b$ | Baseflow | $ks*(SM_2/SM_2^{max})^n$ |
| $Q_{bO}$ | Bucket Overflow | $Q_{bo,i} = \text{MAX}[0,(SM_i - SM_i^{max})]$ |
| $Q_d$ | Overland Flow | $Q_{in}*Ac$ |
|  |  | $Ac = \text{MAX}[0, 1-(1-SM_1/Sm_1^{max})^\beta]$ |
| $ET$ | Evapotranspiration | $ET_1 = PET*(\text{MIN}[SM_1, SM_{T1}]/SM_{T1}^{max}$ |
|  |  | $ET_2 = (PET - ET_1)*\text{MIN}[SM_2, SM_{T2}]/SM_{T2}^{max}]$ |
| $Q$ | Streamflow | $Q_b + Q_{bO} + Q_d$ |

**Table 2.** Functional Forms for Bucket Hydrologic Model Fluxes

In the second step, SNOW-17 is coupled with a bucket hydrologic model based on the FUSE hydrologic model framework Clark et al. (2008). SNOW-17 provides rain and snowmelt inputs to the hydrology model. The hydrologic model also requires a potential evapotranspiration forcing, which is computed using the Hamon formula (Hamon W. R., 1961). The model structure used in this study is the most similar to the VIC/PRMS forms described in Clark et al. (2008). The structure was chosen for

simplicity and to have as few free parameters as possible. There are two state variables, soil moisture in the top and bottom buckets ($SM_1$, $SM_2$) with maximum capacities $SM_1^{max}$ and $SM_2^{max}$. The model flux equations are solved sequentially and time integration is performed with a basic forward Euler method using a daily timestep. The model flux equations are described in Table 2 and illustrated in Figure 2. ET also depends on the fraction of water held in tension storage in the top and bottom layers ($SM_{T1}$ and $SM_{T2}$) which are simply given by $SM_{T1} = fracten*SM$. The summation of the bucket overflow ($Q_{Bo}$),

baseflow ($Q_B$) and direct or overland flow ($Q_d$) is convolved with a routing function kernel to produce a streamflow that can be compared against USGS observations at the gauge site at the ERW outlet (Figure 1).

The posterior model parameters ($\theta$), conditioned on the model structure and observed streamflow data ($d$), are can be expressed using Bayes' rule: $P(\theta|d) \propto P(d|\theta)P(\theta)$, where $\theta$ is the vector of model parameters and $d$ is data. Analytical expressions for the posterior are not possible, so Markov chain monte-carlo sampling methods are used, specifically the DEM-Metropolis





algorithm implemented in the python "PyMC3" library (Salvatier et al., 2016). The model-likelihood function $P(d|\theta)$ is a log-likelihood error function. We further employ a heteroscedastic error model, which states that the model residual is a normally distributed random variable with a standard deviation ($\sigma_t$) that grow linearly with discharge, following Henn et al. (2015) and Thyer et al. (2009). The model is given by $\sigma_t = a_1 Q_t + b_1$, where $Q_t$ is the observed discharge at timestep $t$. The coefficients of the error model are inferred along with model parameters. PyMC3 allows for a number of options for sampling from the poste-

rior distribution. We found that a large number of samples (>15000) and multiple "chains" (independent posterior samplings) led to better sampler convergences as defined by the well-known Gelman-Rubin statistic implemented in PyMC3 (Gelman and Rubin, 1992). Example traceplots from the MCMC sampling are provided in the supplementary material.

     Bayesian parameter inference is performed in two parts. First, climatological or time-invariant parameters related to the sub-surface hydrologic structure (Table 3) are inferred using precipitation and temperature that have been tuned to match ASO and

SNOTEL observations. The assumption is that these conditions represent good approximations of the climatological behavior of the watershed. In the next step, the new posteriors of the time-invariant parameters are set as fixed and meteorological parameters (Table 3) are inferred against discharge independently for each year, providing posterior likelihoods of precipitation conditioned on the model structure and streamflow data for that year.

     While the details of the inference method are numerous, it is important to keep in mind the ultimate goal. We seek a Bayesian

estimate of basin mean precipitation, independent of WRF and gridded precipitation products, that incorporates available high-quality hydrologic information to serve as an additional constraint on precipitation. The code for performing the analysis is posted publicly on github (Rudisill, 2023).

## 3    Results

### 3.1    Comparison against Liu et al. (2017)

First, to explore the fidelity of our WRF model configuration, we make a comparison against WRF v3.4.1 data from the Liu et al. (2017) 4km dataset for water year 2013. This regional climate run covers the entirety of the continental United States and uses ERA-Interim initial and boundary conditions. Data are accessed from the NCAR research data archive (Rasmussen and Liu, 2017). We compute the percent difference between the Liu dataset and this dataset ($(\sum P_{Liu} - \sum P_{Rudisill})/\sum P_{Liu} * 100$) and find that there is generally good agreement between the datasets (Figure 3). Our WRF configuration produces less precipitation

in the center of the domain, but more on the model boundaries likely related to nesting effects or more finely resolved terrain in this dataset (1km vs 4km). The average precipitation for the entire domain is ultimately quite similar (563mm for this study versus 555mm in Liu et al. (2017)). This lends confidence to the overall skill of this particular WRF configuration, and warrants further evaluation against other data sources.





| Parameter | category | Description | Prior | Range | Relevant Reference |
|---|---|---|---|---|---|
| $SM_1^{max}$ | subsurface | unsaturated zone max storage (mm) | fixed | 400 | Clark et al. (2008) |
| $SM_2^{max}$ | subsurface | saturated zone max storage (mm) | uniform | 10—1000 | Clark et al. (2008) |
| ks | subsurface | percolation rate (mm $d^{-1}$) | uniform | 5—200 | Clark et al. (2008) |
| ku | subsurface | baseflow rate (mm $d^{-1}$) | uniform | 5—6000 | Clark et al. (2008) |
| n | subsurface | baseflow exp. (unitless) | uniform | 1—10 | Clark et al. (2008) |
| $\beta$ | subsurface | saturated area exp. (unitless) | uniform | .001—3 | Clark et al. (2008) |
| c | subsurface | percolation exp. (unitless) | normal | 5, .1 | Clark et al. (2008) |
| fracten | subsurface | field capacity fraction (unitless) | uniform | 0.1—.95 | Clark et al. (2008) |
| $\tau$ | streamflow | routing timedelay (days) | uniform | .01, 3.5 | Clark et al. (2008) |
| $a_1$ | streamflow | error model coefficient | normal | 0, .15 | Thyer et al. (2009) |
| $b_1$ | streamflow | error model coefficient | halfnormal | 0, .5 | Thyer et al. (2009) |
| OPG | atmospheric | precip gradient (m$^{-1}$) | uniform | .0005—.007 | Kirchner et al. (2014), Henn et al. (2015), |
| m | atmospheric | precip error multiplier (unitless) | normal | 0, .1 | Henn et al. (2015) |
| $\zeta$ | atmospheric | Equation 2 (m) | uniform | 3000—3800 | - |
| $\alpha$ | atmospheric | Equation 2 (unitless) | normal | .5—1.5 | - |
| t-lapse | atmospheric | temperature lapse rate (° C m$^{-1}$) | normal | -.004,.005 | Lute and Abatzoglou (2021) |
| t-bias | atmospheric | temperature sensor bias (° C) | uniform | -2., 2. | Oyler et al. (2015) - |

**Table 3.** Model parameter prior values and probability distribution used in the precipitation inference method. Ranges refer to the min/max of the uniform distribution or the mean/standard deviation of a normal distribution.

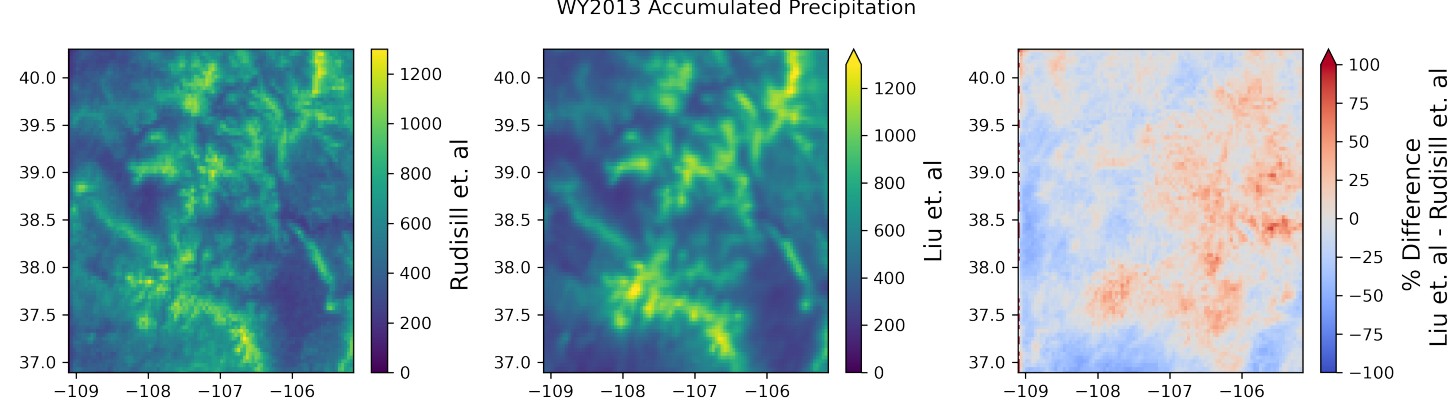

**Figure 3.** Total accumulated model precipitation for Water Year 2013 from this study compared against data from the Liu et. al 2017 4km CONUS WRF dataset



## 3.2 Seasonal Precipitation Accumulations Compared Against SNOTEL

WRF precipitation from the 34 year period is compared against corresponding SNOTEL grid-cells shown in Figure 4. Comparing WRF precipitation against SNOTEL observations demonstrates that WRF captures important features of atmospheric water-delivery throughout the region. The analysis is divided into two rough categories, the "cold-season" (October-March) and "warm-season" (April-September) which are intended to roughly demarcate winter stratiform and summer-time convective precipitation regimes. Analyzing the monthly averages of integrated vapor transport and 500 hPa wind directions shows that wind and moisture overwhelmingly come from the West during the cold-season (October - April) and from the West-South-West during the remainder of the year (not shown). The percent errors in water-year total precipitation, expressed as

$$\% \text{ Error} = \frac{\sum P_{WRF} - \sum P_{SNOTEL}}{\sum P_{SNOTEL}} * 100 \tag{3}$$

are also examined. There is no immediately apparent trend in location of SNOTEL site with respect to elevation or topography and error characteristics. The worst performing site is Brumley (369), located on the lee-side of a mountain ridge, where WRF overpredicts precipitation consistently throughout the study period. Interestingly, sites located only a few kilometers away on the windward side of the range are well predicted. While the correlations are similar between seasons, the errors in precipitation accumulation are not evenly distributed across the water-year. During the warm-season, WRF is wetter than SNOTEL sites for sixteen of the twenty-three SNOTEL sites, with an average accumulated precipitation percent-bias of 10.4%. The cold-season percent error averaged across all years and SNOTEL locations is, remarkably, .264% but with a 10.1% standard deviation. Comparing one-week rolling mean time series of WRF averaged across all SNOTEL locations, compared with the average SNOTEL precipitation, shows good correlation ($R^2$ of .85 and .88 for the warm and cold-season, respectively). The relationship between binning-window (daily-monthly) and correlation was also examined, and found that correlations were low at the daily increments, but tended to flatten beyond averaging window-lengths greater than three to four days. There is no clear relationship between elevation and precipitation error.

## 3.3 Regional Comparison of WRF and Gridded Datasets

Three gridded precipitation datasets, namely the PRISM (4km or $\sim 1/25$ °), Livneh (1/16°), and Newman (1/8°) products are compared in Figure 5 for water years between 1987 and 2012, as this is the time frame for which all data products are available. WRF is treated as the reference dataset, and the difference and percent error between each product are compared at the resolution of the coarsest product. For ease of comparison we also bi-linearly interpolate the curvilinear 1km regular WRF grid onto a 1/100 ° regular lat-lon grid. The differences in topography between the high-resolution WRF grid and the coarsest grid (1/8°) are also shown. The comparison of water-year averaged precipitation (1987-2012) shows significant differences between WRF and the gridded datasets, on the order of plus or minus 25% and up to 400 mm in some regions. The differences among the gridded datasets are lesser, in particular between PRISM and Livneh. In general WRF shows systematically less precipitation in the San Juan, Front Range, and Elk mountains but sometimes more in the valleys and plateaus between ranges.



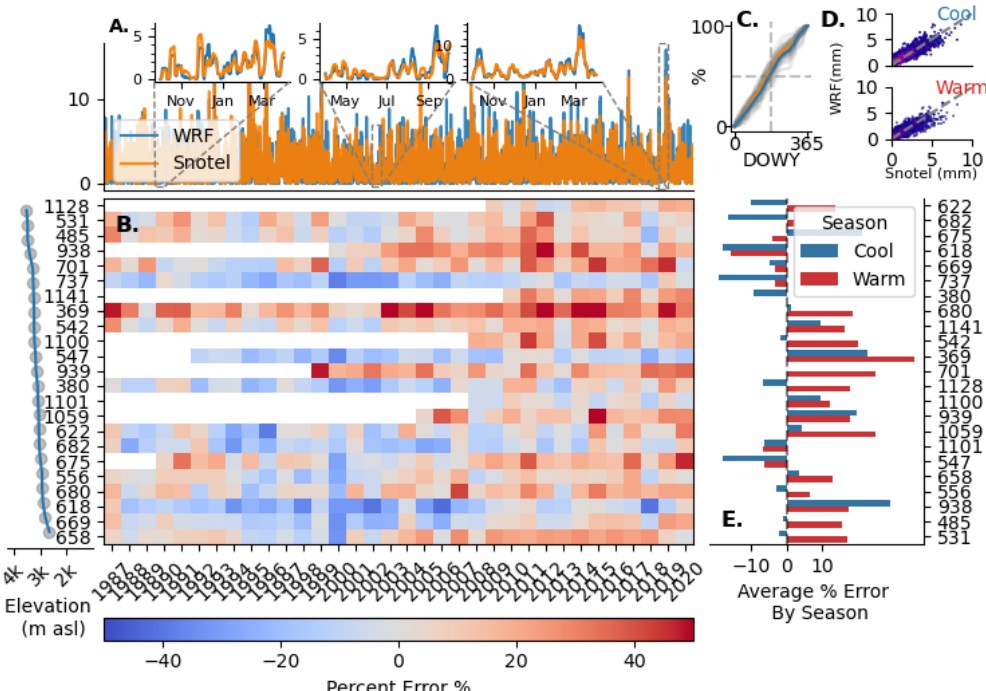

**Figure 4.** Error characteristics of 24 SNOTEL sites compared against corresponding WRF grid-cells, 1987-2020. A) One-week rolling mean timeseries of average SNOTEL (orange) precipitation compared against WRF (blue) B) Annual precipitation percent error (SNOTEL as reference) for each site, C) Average timing of water delivery (%) as a function of day of water-year for WRF (blue) and SNOTEL (orange), D) Correlations between one-week accumulated precipitation WRF/SNOTEL for Warm season (bottom) and Cool season (Top), E) Average precipitation percent errors by season

PRISM shows a noticeable region of enhanced precipitation in the middle of the Elk mountain range (not far from the ERW) that is not as pronounced in the other geostatistical products. The differences between WRF and Newman are broadly similar to the other products, though WRF shows more precipitation in the southwest corner of the domain around the Sangre De Cristo mountains, whereas the opposite is true when compared against the other products. The mountain range is narrow and not well captured by the coarse scale of the topography at the 1/8° resolution.

Comparing precipitation as a function of terrain elevation (Figure 6) shows that WRF typically has less precipitation at the highest elevations (greater than 3350 m) compared to PRISM, and that the two datasets disagree the most in regions that are poorly sampled by SNOTEL locations (the SNOTEL maximum elevation is 3500 meters). Livneh and Newman were also examined as functions of height but were ultimately very similar to the PRISM pattern and are not shown. PRISM has a more skewed distribution at higher elevations in addition to higher maxima than WRF. Both datasets show a clear rain-shadowing

effect between 2250-2400 meters, corresponding with the region to the East of the San Juan mountains in the South East corner of the domain, though PRISM is drier than WRF (Figure 5).



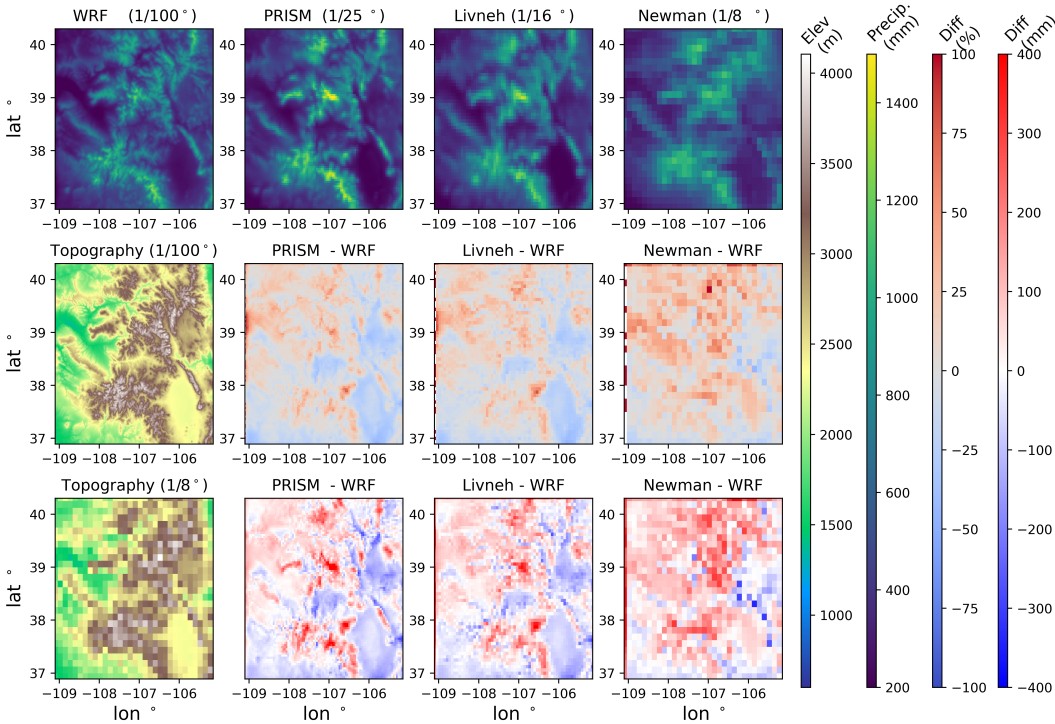

**Figure 5.** Top row: Spatial averages of 1987-2011 annual precipitation for WRF, PRISM, Newman, and Livneh datasets for the entirety of the inner WRF model domain at each respective data resolution. Topography at WRF and coarsest product resolution are also shown (left column, 2nd and 3rd rows). The difference and percent difference between each product and WRF are shown in the second and third rows.

## 3.4 ERW Precipitation Analysis and Reconstruction

To better understand the patterns and processes at the watershed scale, we examine precipitation estimates across the ERW. Unlike the regional comparisons, each product has been bi-linearly interpolated to the 1x1 km WRF grid. Figure 7 shows the clear effect of resolution on precipitation reconstruction in the ERW, as the Newman and Livneh datasets have much coarser textures than the others and do not reflect the underlying terrain very well. However the ERW-mean precipitation between Livneh and Newman are ultimately quite similar to PRISM, so consequently only PRISM and WRF are compared subsequently. WRF shows significantly less precipitation than any other other products, particularly in the mid 1990s to early 2000s.

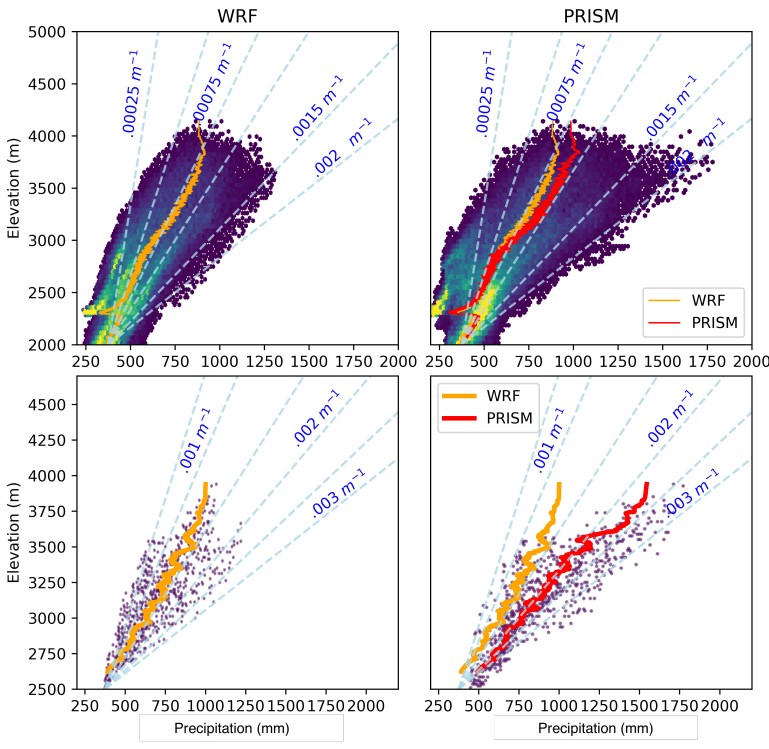

**Figure 6.** Average annual precipitation versus elevation for WRF and PRISM, for the entire WRF domain (top) and the ERW (bottom). Rolling means (solid lines) are shown. For comparison, constant OPG lines from Equation 1 using a starting value of 300 mm at 2500 m of elevation are also plotted.

In order to better investigate some of the spatial characteristics of precipitation and relationships with terrain, we examine WRF and PRISM reconstructions for the warm and cold seasons averaged between water-years 1987-2020. To do so, we define an "enhancement factor" for each grid cell, which is simply each grid cell normalized by the watershed mean value:

$$EF_{i,j} = \frac{P_{i,j}}{\overline{P}} \tag{4}$$

Which is simply the ratio of the accumulated precipitation in each grid cell $i,j$ to the $m$ by $n$ points averaged across the
watershed.

   Both datasets show that more precipitation accumulates during the Cold season on average. In many years, the mountain slopes on the windward side (West; left of the figure) receives more precipitation in WRF relative to PRISM (Figure 8), despite





**Figure 7.** ERW Annual Precipitation by Water Year, 1987-2020 for WRF and PRISM, Livenh, and Newman precipitation products

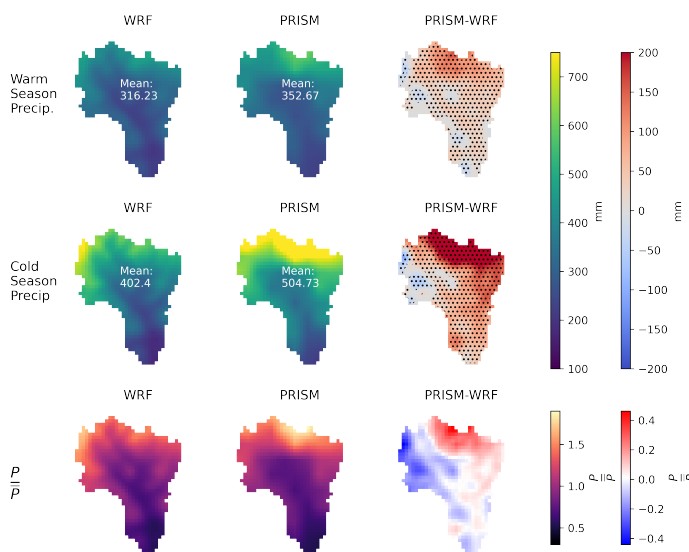

**Figure 8.** Average Cold-Season (October-April), Warm-Season (April-October), and average annual precipitation in addition to the enhancement factor (precipitation normalized by the ERW mean precipitation) for WRF and PRISM across the ERW. Grey dots indicate regions with a difference that is statistically significant at the 95% confidence level using a two-sided t-test.

having an overall lower precipitation. A two-sided t-test shows that the difference in the mean precipitation between the two different datasets is statistically significant for the majority of the grid points with the exception of a few small regions.

WRF has a much higher enhancement factor on the windward East side compared to PRISM, which has a very high enhancement factor and very high positive bias relative to WRF in the Northwest. There does appear to be a significant downward shift in PRISM basin-mean precipitation in the final decade of the simulation, the causes of which are discussed in the next section. PRISM generally has a higher precipitation-elevation gradient compared to WRF, for both the ERW and the entire WRF domain (Figure 6). The averaged elevation/precipitation relationship is most similar at low-to mid elevations and deviates most

strongly at the highest elevations.

### 3.4.1   Comparisons against Bayesian Precipitation Reconstructions

In order to better understand some of the discrepancies between WRF and gridded datasets, a Bayesian precipitation inference method adapted from (Henn et al., 2015) is adapted to examine basin-mean precipitation in the ERW. We analyze water-years 1990-2020 as this is the time period in which high-quality USGS observations are available. Again, this method cannot isolate

spatial precipitation patterns (which are significant; Figure 8), only the basin-mean precipitation. This is nonetheless useful as





the differences in the mean are large ($\sim$ 150mm). A three-part inference process is applied. First, SNOW-17 parameters are calibrated for the Butte SNOTEL (381; Figure 1) site, including a precipitation undercatch term using a standard (non-bayesian) error minimization algorithm ($P_{bias}$, Equation 1). Given the relatively short number of model-years, we choose to use the entire timeseries for calibration as opposed to splitting into calibration and validation periods, which is common but not always optimal in hydrologic contexts (Arsenault et al., 2018). Ad-hoc splitting tests were performed and there was not much impact on the final calibrated values. After calibration, SNOW-17 can very well capture the dynamics of snow accumulation and melt at the Butte SNOTEL site (see supplementary material). Next, the orographic precipitation gradient ("OPG"), temperature lapse rates, and precipitation gradient cutoff terms are calibrated against two-years of Airborne Snow Observatory SWE products. The average ASO SWE from each elevation band is computed and compared against SNOW-17 run in elevation bands with the updated precipitation and temperature inputs. Water year 2018 and 2019 are low and high precipitation years, respectively, with approximately peak SWE values of greater than 2000 mm in 2019 and approximately 1000 mm in 2018 (Figure 9). This is fortuitous, as these years represent high and low extremes, and are thus good for bracketing the long-term average behavior. Aggregating SWE with respect to elevation bins shows a remarkably consistent increase in SWE with elevation, after which SWE values tend to decline. A similar pattern is found in the Tuolumne basin in California (Henn et al. (2016), Figure 4). Tuning the OPG-gradient slope-break ($\zeta$) and decrease-rate ($\alpha$) in Equation 2 allows for fitting the observed ASO SWE curves quite well. It is worth noting that the precipitation reduction term ultimately represents a small fraction of the total watershed area (less than 10%). The calibrated OPG parameter is ultimately close to the initial guess of .002 m$^{-1}$. This guess happens to be very close to the PRISM precipitation/terrain relationship (Figure 6). The temporal evolution of SWE in the corresponding elevation bin also closely matches that of the SNOTEL site (Figure 9.c), demonstrating that this simple model can still capture key features of snow accumulation and ablation well.

Following the calibration of the SNOW-17 model parameters, the time-invariant subsurface and error model parameters are inferred using the Bayesian MCMC method from 1990—2020. The specific parameters are the subsurface parameters listed in Table 3. These parameters are related to geomorphic features of the watershed (soil depth, hydraulic conductivity, etc) and are hypothesized to be largely unchanging during the study period. Extensive testing of the Bayesian inference approach found sampler convergence criteria were satisfactorily met after treating unsaturated zone maximum soil moisture storage as fixed. A value of 400 mm was chosen, reflecting approximately 1 meter on average of sandy-loam soil. Inference was performed with different fixed values of soil water storage, and found there was ultimately little difference in the inferred values of other parameters. The baseline model skill is high at this stage, with average root mean squared error of .65 mm prior to inferring precipitation forcing errors, suggesting that the model structure is a good approximation of the watershed dynamics. The MCMC traceplots and the prior and posterior time invariant model parameters are shown in the supplementary material.

After inferring the time-invariant subsurface model parameters, the meteorologic adjustment parameters are inferred on an annual basis against streamflow with the mean posterior time-invariant subsurface model parameters inserted in the model and treated as fixed. This includes the precipitation error multipliers ($m$), OPG, OPG gradient slope-break ($\zeta$) elevation, and sensor temperature bias ("atmospheric" parameters in Table 3). The convergence of the posterior sampling for each case is evaluated using the Gelman-Rubin (Gelman and Rubin, 1992) statistic.

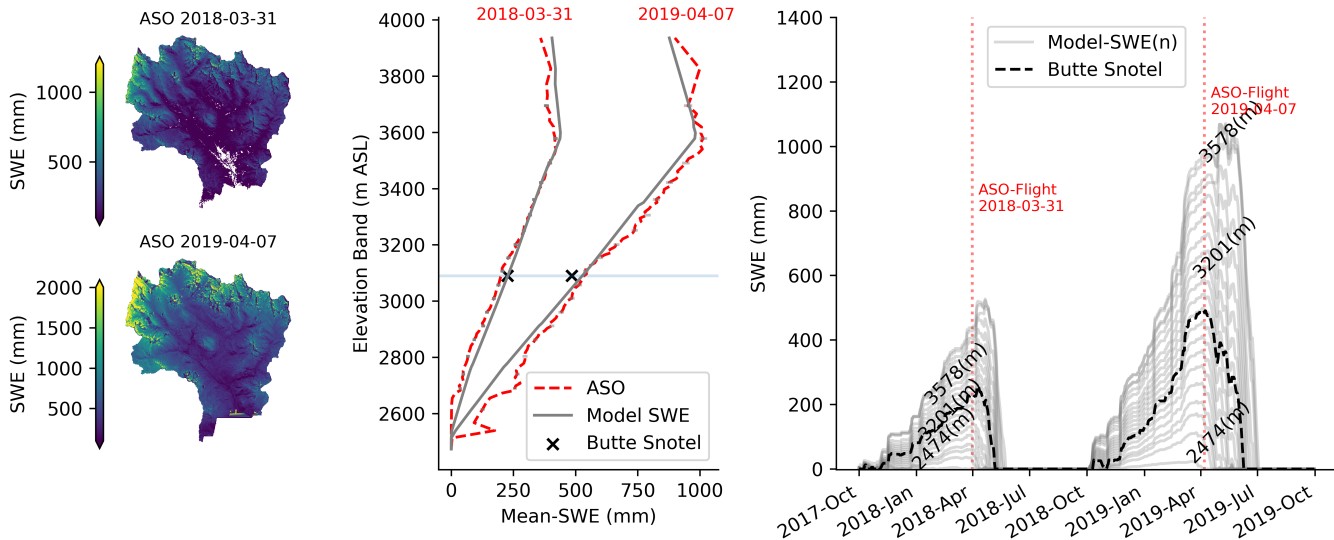

**Figure 9.** Left: The Airborne Snow Observatory SWE maps over the ERW employed in this study. Center: Basin-average elevation versus average SWE for ASO and calibrated SNOW-17 at ASO-flight dates. Right: Timeseries of calibrated SNOW-17 by elevation band (n=100).

Figure 10 shows the priors and posterior parameter distributions for each water year. Interestingly there are one or two years where the inferred temperature lapse rate is positive. This isn't completely nonphysical as cold air pools are very common in this watershed particularly in the winter. We can now examine both the posterior model solutions for streamflow and the posterior probabilities of basin-mean precipitation for each year. A gaussian kernel density estimate with a bandwidth of 2 mm

is used to sample the posterior from each year and produce the plots in Figure 11, showing the comparison against inferred, WRF, and PRISM basin-mean precipitation. There is considerable year-to-year variability in the relationship between the three estimates, but in general the inference approach in between the WRF and PRISM estimates.

Averaged across all years, the Bayesian inference method estimates 761 mm of precipitation, WRF 725 mm, and PRISM 862 mm. Consequently WRF is slightly closer to the Bayesian inference reconstruction. For comparison, the average runoff

from the ERW at the gauge outlet is 370 mm, so all estimates have a reasonable ratio of runoff to precipitation. The timing and magnitudes of modeled streamflow compare well against the observed streamflow, lending confidence in the appropriateness of the model structure and parameter ranges (Figure 12). Averaged across all years, the precipitation error multiplier has a mean close to 1 and OPG parameters are again close to the initial guess of .002 m$^{-1}$ (not shown). The posterior solution model skill-scores are high indicating that the model well captures the observed The entire discharge record has a kling-gupta-efficiencies

(Gupta et al., 2009) greater than .9 (with 1 being a perfect match) and a root-mean square difference (RMSD) of .524 $mmd^{-1}$, which is an improvement from before.

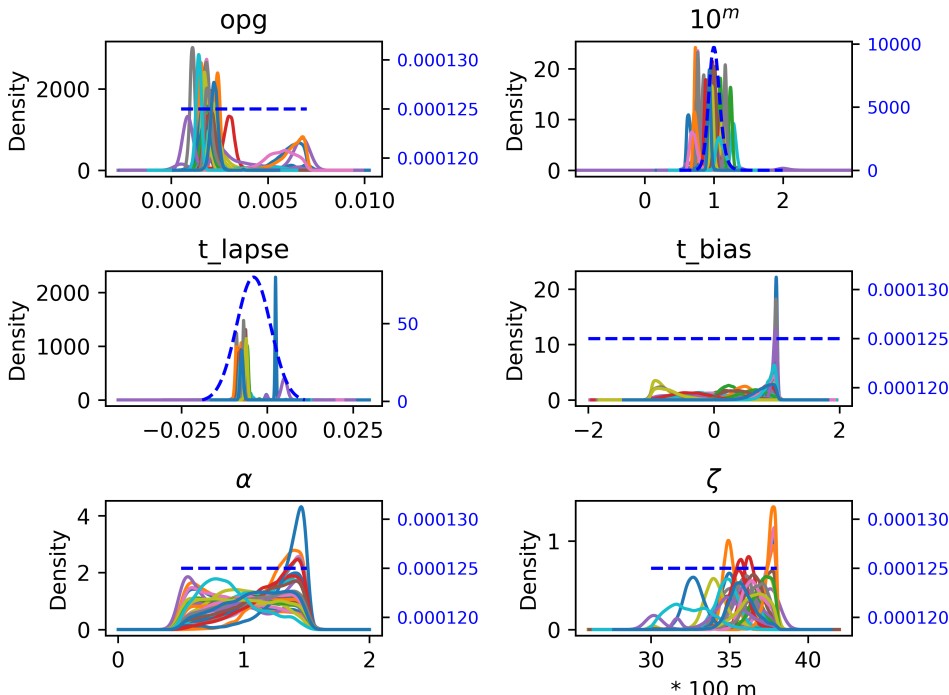

**Figure 10.** Posterior and prior (blue dashed line) meteorological parameters inferred for each water year, 1990—2020

.

Further inspection of the data shows that the three estimates (WRF, PRISM, Inferred) tend to become approximately more alike after the water year 2010. This change in behavior is almost certainly due to the addition of the Upper Taylor SNOTEL in water-year 2010, immediately to the East of the ERW (Figure 1), as PRISM uses this data source. Of the three SNOTEL

sites closest to the ERW, Butte receives the least precipitation, Schofield the most (almost twice that of Butte), and the Taylor typically a value in between the two. This relationship is illustrated in Figure 13. The PRISM basin-mean precipitation is a fairly constant +200mm (standard deviation ∼ 60mm) relative to the precipitation received at the Butte SNOTEL site, and the difference between the two values is significantly reduced after water year 2010 with the addition of the Upper Taylor site.

## 4   Discussion

### 4.1   Comparisons against other Regional Climate Reconstructions

Many other studies have evaluated WRF against SNOTEL and gridded precipitation datasets for the Colorado Rockies, though few have presented analyses for simulations spanning three decades. Rasmussen et al. (2011) reports similar performance metrics between SNOTEL stations and WRF for a limited number of winter seasons. Jing et al. (2017) likewise found that



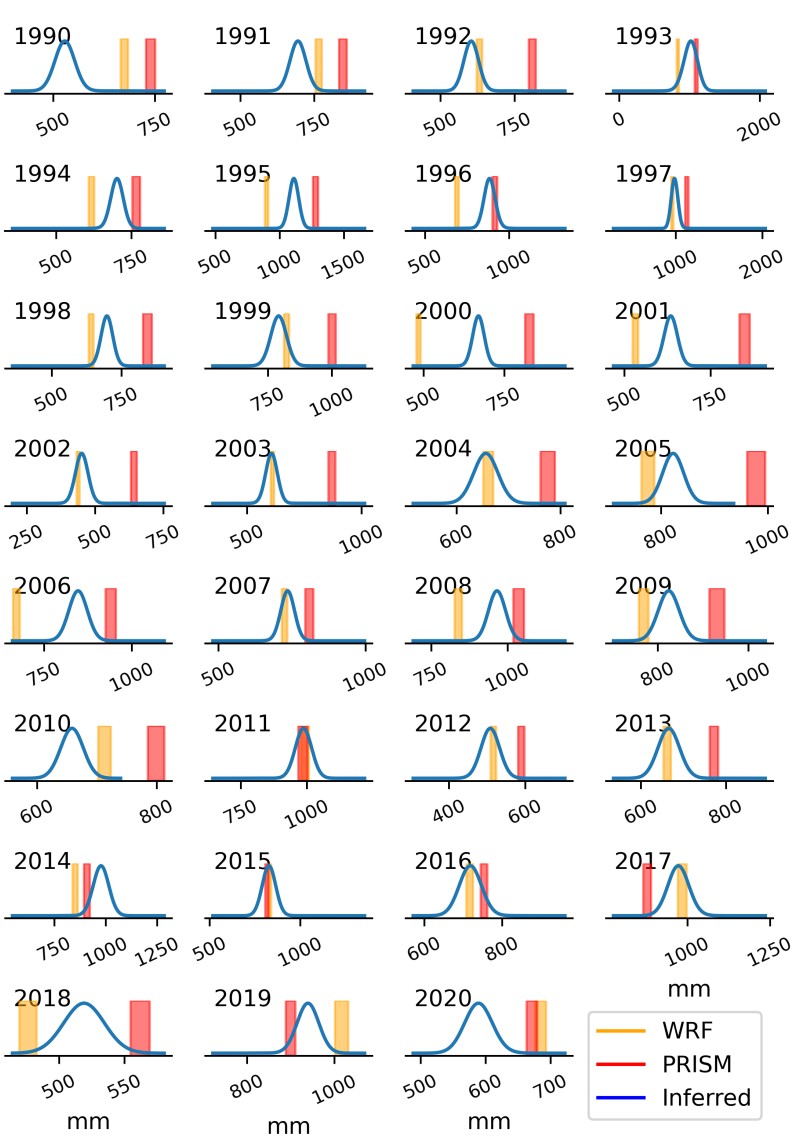

**Figure 11.** ERW Basin Mean Precipitation for water years between 1990-2020 from three different estimates. WRF, PRISM and the posterior probability distribution from the Bayesian MCMC inference method. The widths of the PRISM and WRF bars are the standard error of the respective mean

.





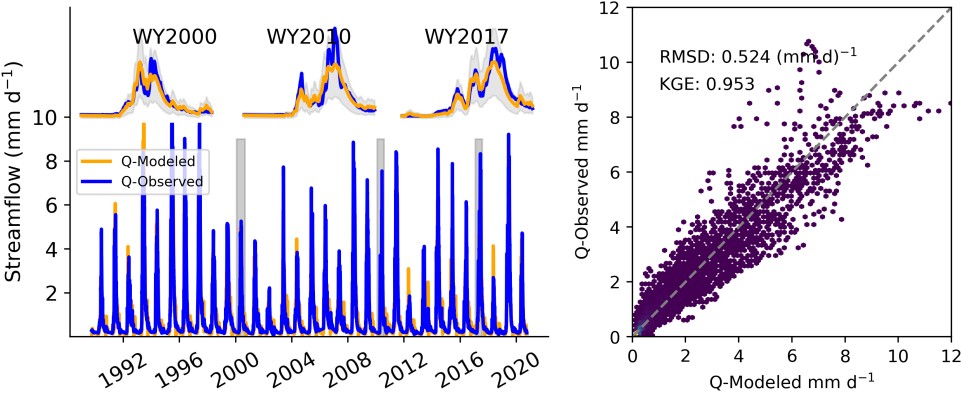

**Figure 12.** Observed streamflow compared against posterior model solutions of streamflow. The shaded region shows the 75% model confidence intervals. Example water years are plotted on top to highlight details in the simulated hydrographs. The right panel shows the daily correlation between model and observed streamflow.

winter-time precipitation accumulations compared against a number of SNOTEL sites was less than 15% using a 2km WRF

configuration with NARR boundary conditions. The absolute biases and percent-differences between WRF and PRISM from this study are similar both the Jing et al. (2017) 2km WRF and 4km WRF results from Liu et al. (2017), both presented in Figure 7 from Jing et al. (2017). Similar performance against SNOTEL is also reported in Gutmann et al. (2012) who examined winter-only precipitation using a 2km WRF configuration again using the NARR boundary conditions. The similarity of results is interesting since there are differences in resolutions, nesting configurations, model code versions, boundary conditions,

microphysical schemes and other options among all of these models. Xu et al. (2022) used the same physics options and boundary conditions as those used in this study, and found that WRF performed better for simulating near surface climate than several other configurations for WY2019 across the ERW.

Differences in error behavior between cold and warm seasons can likely be attributed to precipitation generating regimes. WRF has a lower weekly correlation with SNOTEL stations (lower $R^2$ value) and higher percent errors. Additional analysis

shows that the warm season bias is related to an excessive number of wet days relative to SNOTEL during the warm season. Jing et al. (2017) likewise found that WRF skill decreased in April, concurrent with an increase in convective available potential energy in the atmosphere. Surface heating tends to increase during the warmer months leading to convective instabilities and localized precipitation, compared to more uniform stratiform precipitation during the winter (Dai, 2006). At the same time, the nature of errors at the gauge locations can depend on the phase of falling hydrometeors (rain versus snow), as snowflakes

have slower fall speeds and are more subject to undercatch during stronger winds (Goodison et al., 1998; Harpold et al., 2017). It is even more interesting that cold-season errors are lower than warm-season errors, as the gauge errors are expected to be higher in the cold-season when more precipitation falls as snow. Some studies have attempted to account for gauge undercatch at SNOTEL sites using the co-located snow pillow SWE measurements and wind observations, but these methods are not

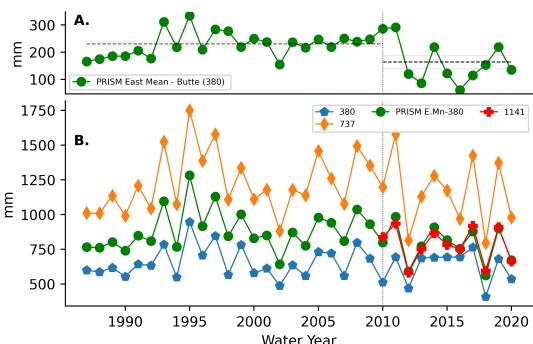

**Figure 13.** ERW annual mean PRISM mean precipitation compared against the three closest SNOTEL sites, Schofield (737; North of ERW), Upper Taylor (1141; East of ERW), and Butte (381; within ERW). Pre/Post 2010 mean differences between Butte and PRISM-mean are plotted with standard errors.

employed here for the precipitation evaluation (Livneh et al., 2013; Sun et al., 2019). However, an undercatch term ($P_{bias}$) for
the Butte SNOTEL was used in the precipitation inference method, but the value was small (< 1 mm), and only two-years out of the thirty examined appeared to show snow-water equivalent values greater than the co-located accumulated precipitation. The overestimation of the number of wet days is well known in both regional and global climate modeling (Maraun et al., 2010; Chen et al., 2021). However, it is difficult to determine the extent of the potential drizzle bias given the relatively low resolution (daily; 2.54 mm) of SNOTEL gauges. NRCS provides recent data with sub-daily frequency, but not for the entirety of
the analysis period. Additional comparisons with other high-resolution gauge data may shed light on some of these questions. Additional experiments may consider indirect data sources (soil moisture, remote sensing) to better understand modeling drizzle days in regions covered by SNOTEL, which account for ∼ 10% of annual WRF precipitation. Users of regional climate model precipitation may consider bias correcting by removing low-precipitation days (Maraun et al., 2010).

### 4.2 Timescales and Data Non-Stationarity

This study illustrates some of the challenges of multi-decadal model evaluations, and highlights the utility of using streamflow observations as additional long-term water balance indicators. The WRF dataset covered 1987-2020, whereas the Livneh and Newman datasets were only compared up until 2012 because of data-coverage. Moreover, only 14 of the 23 regionally relevant SNOTEL sites covered the entire data period. Streamflow, on the other hand, can often have much longer records Tillman et al. (2022). This study demonstrates that streamflow can be a useful data point for evaluating precipitation budgets, given that
comparison gridded precipitation datasets are imperfect. For instance, the step-change in PRISM basin-mean precipitation after approximately water year 2010 (Figure 13) underscores the fact that caution must be taken when analyzing precipitation trends from gridded datasets, as the inclusion of different stations can induce spurious trends. For instance, the annual ERW mean precipitation from PRISM from 1987-2020 shows a negative trend with p=.06, $R^2$=.33, which likely reflects the addition of the





Taylor SNOTEL site. The fact that PRISM more closely matches the WRF and inferred precipitation data (two independent

estimates) in the last decade, after the addition of a nearby gauge, lends more confidence to the WRF modeled precipitation

in the early parts of the simulation. A similar conclusion was also reached in Gutmann et al. (2012). That being said, WRF

still typically underestimates precipitation at the Schofield SNOTEL site (737; Figure 4), so the low bias may carry over to

adjacent regions within ERW. Moreover, the LBC conditions change change from the CFSR (Saha et al., 2010) to the CFSv2

analysis (Saha et al., 2014) after 2011. Regional climate models are ultimately dependent on the boundary conditions (Goergen

and Kollet, 2021), so changes in data assimilation and reanalyses methodologies likely influence the character of the boundary

conditions and the results. The quality and quantity of data assimilation greatly increases after during the late 1990s and early

2000s (Saha et al., 2010) in addition to model structural changes between data products. It is possible that this is the reason that

WRF precipitation tends to transition from low-biasd to high-biased when evaluated against SNOTEL during the late 2000s.

### 4.3  Interpreting Bayesian Precipitation Reconstructions

Combining snow remote sensing, streamflow observations, and parsimonious hydrologic models in a Bayesian framework

allows for creating novel constraints on watershed average precipitation that leverages both the high spatial resolution of

ASO snow data and long-records of streamflow. Henn et al. (2016) likewise used ASO data in a joint-inference method in

the Tuolumne river watershed and found that doing so reduced the dependence of inferred precipitation on hydrologic model

structure, compared with inferring precipitation from streamflow alone. That being said, uncertainties in soil parameters, PET

forcing, and water limitation relationships in the PET/ET relationships do limit the conclusions drawn from the precipitation

inference approach. ET is directly related to inferred precipitation by the hydrologic mass-budget equation, $Q = P - ET - \frac{dS}{dt}$,

so holding Q and $\frac{dS}{dt}$ constant implies that higher annual inferred precipitation requires higher annual ET. In the bucket model

formulation used here, ET is a function of PET forcing and the soil moisture content, given by

$$ET = PET * \frac{\min(SM_{T1}, SM_{T1}^{max})}{SM_{T1}^{max}} \tag{5}$$

where $SM_{T1}^{max}$ is the field capacity given by $fracten * SM_1^{max}$ (soil moisture in the top bucket multiplied by the time-

invariant fraction held in tension). During the long-term hydrologic parameter inference, the $fracten$ parameter consistently

converged towards the limit of 1 (a lower-ET solution). Additional sensitivity experiments show that higher ET solutions (lower

$fracten$) tended to smooth spring and early summer streamflow peaks excessively compared to observations, suggesting some

confidence in the ET solution. For these reasons, it is also unlikely that PET forcing parameterization has significant impact on

the overall water-budgeting, as the soil moisture availability term also has a significant impact on actual ET.

Ryken et al. (2021) deployed eddy-covariance towers in riparian zones in the ERW, and reported annual ET values between

450-550 mm per year. This is interpreted as an upper-limit of basin wide ET, as the flux tower is located in a well-watered

riparian corridor. The posterior ET solutions from the inference method are the same magnitude, and between 350-450mm for

the same time period. At the same time, Carroll et al. (2019) estimated ET for a high-elevation sub-watershed of the Upper

ERW as over 623±50 mm/year, and Tran et al. (2019) found ET values between 431 to 624 mm/year for the entire watershed,


which is the most relevant for this study as it considers the same watershed extent. This is closer to, but still higher than, the average of 388 +/- 50mm that is found from the inference method in this study. It is important to note that the Tran et al. (2019) model study uses PRISM precipitation input, and a different precipitation dataset (such as WRF) may produce a different result, as ET depends on the antecedent precipitation.

Another major assumption is that parameters in the precipitation distribution functions are constant for each season. Efforts could consider inferring precipitation error parameters for individual storm events such as in Le Moine et al. (2015) and Koskela et al. (2012). Moreover, while this study demonstrated that airborne snow lidar products can be incorporated into a Bayesian inference strategy to evaluate orographic precipitation, the potential applications are vast. This study only used one-dimensional SWE versus elevation information as part of the Bayesian inference framework to calibrate the climatological OPG parameter

in Equation 2.3. Cursory analysis shows that the patterns of SWE from the ASO data (Figure 9.A) are very similar to the relative precipitation enhancement from WRF, where the largest values of precipitation/snow are on the windward (Western) ridge the watershed. This region also happens to generally be the furthest from observations, so ASO provides a very unique window into the spatial water-budgets of this region and supports the precipitation enhancement shape as modelled by WRF. This study only used the Bayesian inference approach for the ERW, but similar methods could be applied to all headwaters

basins in the domain without significant anthropogenic disturbances. Modern software packages such as PyMC3 (Salvatier et al., 2016) are well developed infrastructures that make implementing MCMC calculations significantly easier than ad-hoc approaches.

## 5    Conclusions

Summary and Conclusions This study examined 34 years of precipitation from a high-resolution (1x1 km) regional climate

model (WRF v3.8.1). Precipitation is compared against the Liu et al. (2017) dataset, SNOTEL observations, three different precipitation products (PRISM, Livneh, and Newman), and basin-mean precipitation inferred from a Bayesian inference method that uses streamflow and high resolution snow lidar remote sensing data (Painter et al., 2016). The primary goal is to better characterize precipitation biases and error characteristics from this regional climate model simulations, while acknowledging that gridded precipitation estimates are themselves models subject to large uncertainties. We showed that:

– Averaged across 24 SNOTEL stations, WRF has a average .246% percent bias (s=13.63%) during the cold season and an average 10.37% (s=12.79%) percent bias during the warm season, suggesting that cold-season precipitation is better predicted despite being more difficult to accurately observe due to undercatch effects.

  – PRISM/Livneh/Newman show generally similar patterns and disagree with WRF on the order of +/- 20% per year. The largest disagreements are at the highest elevations above the SNOTEL measurement network elevation.

– WRF is slightly closer to a Bayesian reconstruction of basin-mean precipitation, conditioned on streamflow and snow remote sensing, than the PRISM estimate. Combining multiple sources of hydrologic information in a Bayesian framework proves to be a useful data-point for model evaluation.





- – WRF shows a spatial pattern of precipitation enhancement in the ERW that more closely matches patterns found in high resolution snow remote sensing data (ASO) than that of PRISM

- – Geostatistical, grid-based based products are influenced by spatial density changes over time and should be treated with care for long-term model evaluations.

Many of the insights about uncertainties in mountain precipitation have been made in other studies (Lundquist et al., 2019; Henn et al., 2018; Daly et al., 2008). While they serve a clear purpose for model development, the limitations of gridded precipitation data in mountain terrain should be carefully considered, particularly when high resolution modeling strategies 460 (enhanced parameterizations, grid spacings, etc) are employed. Using high quality, non traditional hydrologic indicators (such as snow remote sensing) should be a priority for regional climate simulation development.





*Data availability.* A subset of the WRF RCM dataset covering the ERW are available on the ESS-Dive repository https://data.ess-dive.lbl.gov/datasets/doi:10.15485/1845448. We are working on making the entire WRF dataset publicly available, researchers are encouraged to reach out about data availability. USGS streamflow data were downloaded using the https://github.com/DOI-USGS/dataRetrieval package.
NRCS SNOTEL data are downloaded from the web using https://github.com/bsu-wrudisill/NRCSSnotel-Downloader. Livneh precipitation data is available from https://psl.noaa.gov/data/gridded/data.livneh.html, Newman from https://www.earthsystemgrid.org/dataset/gridded_precip_and_temp.html, and PRISM from https://prism.oregonstate.edu/. ASO data are downloaded from the National Snow and Ice Data center from https://nsidc.org/data/aso/data. CFSR and CFSv2 are downloaded from https://www.ncei.noaa.gov/thredds/catalog.html

*Code and data availability.* The codes for performing the analysis, Bayesian reconstruction and hydrologic bucket model, and creating the figures are publicly available on github: Rudisill (2023). A python wrapper for running the WRF model on HPC systems was developed for this project is available on github https://github.com/bsu-wrudisill/WRF-Run.

*Author contributions.* W. Rudisill ran the WRF simulations, compiled data, performed the analysis, developed the Bayesian reconstruction methodology, and wrote the manuscript. A. Flores provided funding and developed the WRF configuration options. R. Carroll provided expertise on the hydrology and climate of the ERW and provided edits for the manuscript.

*Competing interests.* We declare no competing interests.

*Acknowledgements.* Rudisill and Flores acknowledge funding from DOE BER grant DOE:DE-SC0019222. This research made use of Idaho National Laboratory computing resources which are supported by the Office of Nuclear Energy of the U.S. Department of Energy and the Nuclear Science User Facilities under Contract No. DE-AC07-05ID14517. We also acknowledge high-performance computing support of the R2 compute cluster (DOI: 10.18122/B2S41H) provided by Boise State University's Research Computing Department.



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
