# Peer review of "Evaluating Three Decades of Precipitation in the Upper Colorado River Basin from a High-Resolution Regional Climate Model"

_Geoscientific Model Development, 2023_

## Author Response (AR1)

**Responses to the Open Discussion Period**

**RC1**

*Comments on "Evaluating three decades of precipitation in the upper Colorado River basin from a high-resolution regional climate model", by Rudisill et al., submitted to Geoscientific Model Development, for possible publication.*

*Accurate precipitation estimate over complex terrains remains a challenge and hot research topic in hydrology communities. The present study demonstrates the utility of high-resolution regional climate model simulations in leveraging both spatial and temporal precipitation variability over the Upper Colorado River basin. They carry out multi-scale evaluation of simulated precipitation fields against in-situ observations and various gridded products (reanalysis, reconstructed). Their results could provide important implications for future precipitation validation studies in complex terrains, and offer a good dataset for hydrology studies in this particular region. The manuscript is comprehensive and overall well-written.*

*A major concern of mine is since the simulations are implemented in hourly scale, I would suggest comparisons at hourly scale should be carried out as well. The authors emphasize the utility of the hourly rainfall products, but nowhere in the text that we can see how the model is compared against in-situ observations, for instance, does the model capture the daily rainfall cycle well? How about the spectrum of hourly rainfall intensity, etc.? This is missing from the manuscript, but is needed for potential users.*

*A minor concern of mine is how are the physics options determined, do the authors test other combinations? We know rainfall simulation is particularly sensitive to a couple of parameterization schemes, e.g., microphysics schemes, planetary boundary layer scheme.*

*The presentation of the manuscript should be improved. The current version seems lengthy. Some paragraphs could not be repeated, for instance, lines 60-80, and also some details in section 2.3 and 4.3 can be presented only once. I also spotted some typos or incomplete sentence, but would suggest the authors to check throughout the manuscript. For instance, line 6, line 31, line 393, line 439.*

Response

Thank you for your review of the manuscript. We have noted some of the grammatical mistakes that you found and fixed them, in addition to others we found. There were a few cases where there were extra parentheses in the in-line citations and we fixed those. We also had many instances where there was no space between a number and the unit abbreviation. We also shortened some of the introduction and simplified/consolidated some of the text so that discussion was not duplicated. We moved one sentence introducing the ERW, removed some of the duplicate discussion of SNOTEL, and also removed some of the extraneous discussion about the inference model parameterizations in section 4.3.

1) Physics combinations--We currently have another manuscript in review in the Journal of Hydrometeorology where we have tested other physics combinations, namely microphysics. We also would like to refer the reviewer to the following study:

Xu, Z., E. R. Siirila-Woodburn, A. M. Rhoades, and D. Feldman, 2022: Sensitivities of subgrid-scale physics schemes, meteorological forcing, and topographic radiation in atmosphere-through-bedrock integrated process models: A case study in the Upper Colorado River Basin. *EGUsphere*, 1–29.

In this study, the same physics options used in this study (referred to as "BSU" in their study, are compared against several others). We mention this in passing in line 357, but we have reworded this sentence to make the significance more clear. They found the best performance of this physics combination for simulating meteorology in the ERW. The new lines are around line 354 of the revised manuscript.

2) Sub-daily evaluations: We agree that subdaily evaluations of precipitation are particularly interesting and there is likely much to explore for this topic. We mention this in line 61. Unfortunately there are not many good data products that provide sub-daily precipitation data for model evaluation in the high-elevation regions that we are primarily interested in for this study. PRISM is a daily product, for instance, and likewise NRCS SNOTEL data from historical periods is usually only available at daily increments. We mention the latter fact in line 374-375.

Moreover, the early drafts of the manuscript did include a plot showing analysis of precipitation intensity at daily scales compared to SNOTEL observations, but we decided that this was too much for an already large manuscript that mostly focuses on seasonal accumulations. We reference some of these results, however. In line 370, we say *"Additional analysis shows that the warm season bias is related to an excessive number of wet days relative to SNOTEL during the warm season"*.

**RC2**

The subject is relevant and the manuscript has been written well. I think that the paper can be considered for publication after a minor revision.

COMMENTS

- Fig.10 is not clear. Most acronyms have not been explained in the caption or on the graph. Each figure must be understandable individually. Also, add explanations for the plotted lines. Are each color for each year? If yes, explain it in the caption.

- Fig.12: The streamflow values are presented in mm/day which is not a commonly used unit for water flow. I suggest to use any "volume/time" unit instead of "length/time". Also, on the right plot, "-1" in "RMSD: 0.524…" has to be moved into the parenthesis.

- It is suggested that in the discussion section, add even few explanations about global gridded precipitation data and refer to relevant papers (e.g., in this paper 15 global gridded precipitation data have been investigated, 10.1016/j.agwat.2021.107222). These global gridded precipitation data can be applied for carrying out similar studies outside of the US.

**Response**

Thank you for your comments, we think they have significantly improved the manuscript.

We have added description to the caption of Figure 10 in the updated version of the manuscript. Each of the colored lines is a different year as you suggest.

We did consider which way to present the units for Figure 12, and agree that volume per time is more typical. However, for the purposes of this study, we prefer to use the units mm/d (daily streamflow normalized by watershed area) as it is more directly comparable to other water balance units such as areal averaged precipitation, which are reported in length units. So integrating under the curve yields annual streamflow in units of length. We also moved the text such that the curves are more legible.

We agree that discussion of global precipitation datasets are a valuable point of discussion and have added a line in the updated version of the manuscript. We provided a reference to the paper that you have referred to, as well as referencing the "WorldClim" dataset which has a global precipitation product. It would be interesting and a worthy task to compare globally spanning gridded precipitation datasets in addition to those that are used here, but we think it is beyond the scope of the current research.

---

## Author Response (AR2)

*The authors have well addressed the reviewers comments. After a second review, there is a minor clarification needed as raised by one of the reviewers. Please justify why only the water year of 2013 is chosen to compare against Liu et al. dataset, and specifically if notable improvement over the Liu et al. was observed by comparing against in-situ observations. After making this clarification, the manuscript can be accepted for publication.*

We selected the 2013 dataset (WY 2013) for specific reasons. Firstly, the length of our study precluded a comprehensive evaluation of the entire Liu et al. dataset (which spans the 2000 to 2013). While such an analysis might be informative, it exceeded the scope of our present study, whose goals were to evaluate the model versus observation based products. 2013 was also selected as it is a somewhat typical year in terms of precipitation (neither anomalously high nor low, as shown in Figure 4). Lastly, from a practical standpoint, 2013 is the most recent year of the Liu dataset, so it is the most pertinent point of comparison for researchers utilizing our dataset in investigations covering the last decade.

While we did not perform a direct skill comparison (i.e., model vs. observations), between Liu et al. and our 2013 dataset, we do find that the performance statistics reported in Liu et al. paper are similar to what we report. Both studies indicate higher precipitation skill during winter compared to summer (Figure 9 of their paper), which matches our results. Regarding bias, Liu et al. reported a small negative bias of -2% in average precipitation across all SNOTEL sites throughout the western United States. In contrast, we report a slightly lower bias of 0.25% (as indicated in line 445) when evaluated against the SNOTEL sites considered in our study. However, a more extensive analysis would be required to determine which model options were more skillful and in what circumstances.

The evaluation of precipitation performance in convection-permitting model studies holds significant scientific relevance, particularly as these studies become more common. We contend that a future investigation, aimed at intercomparing publicly available datasets, including ours, the Liu et al. 2017 dataset, the recently introduced CONUS404 dataset**, and others, could offer valuable insights. Nevertheless, such an endeavor falls beyond the scope of the current manuscript.

*\*\*Rasmussen, R. M., and Coauthors, 2023: CONUS404: The NCAR-USGS 4-km long-term regional hydroclimate reanalysis over the CONUS. Bull. Am. Meteorol. Soc., **-1**,* [https://doi.org/10.1175/BAMS-D-21-0326.1](https://doi.org/10.1175/BAMS-D-21-0326.1).